# Telomere-to-telomere gapless chromosomes of banana using nanopore sequencing

Caroline Belser [1,6], Franc-Christophe Baurens[2,3,6], Benjamin Noel [1], Guillaume Martin [2,3], Corinne Cruaud[4], Benjamin Istace [1], Nabila Yahiaoui[2,3], Karine Labadie [4], Eva Hřibová[5], Jaroslav Doležel [5], Arnaud Lemainque[4], Patrick Wincker [1], Angélique D'Hont[2,3] & Jean-Marc Aury [1✉]

Long-read technologies hold the promise to obtain more complete genome assemblies and to make them easier. Coupled with long-range technologies, they can reveal the architecture of complex regions, like centromeres or rDNA clusters. These technologies also make it possible to know the complete organization of chromosomes, which remained complicated before even when using genetic maps. However, generating a gapless and telomere-to-telomere assembly is still not trivial, and requires a combination of several technologies and the choice of suitable software. Here, we report a chromosome-scale assembly of a banana genome (*Musa acuminata*) generated using Oxford Nanopore long-reads. We generated a genome coverage of 177X from a single PromethION flowcell with near 17X with reads longer than 75 kbp. From the 11 chromosomes, 5 were entirely reconstructed in a single contig from telomere to telomere, revealing for the first time the content of complex regions like centromeres or clusters of paralogous genes.

[1] Génomique Métabolique, Genoscope, Institut François Jacob, CEA, CNRS, Univ Evry, Université Paris-Saclay, Evry, France. [2] CIRAD, UMR AGAP Institut, Montpellier, France. [3] UMR AGAP Institut, Univ Montpellier, CIRAD, INRAE, Institut Agro, Montpellier, France. [4] Commissariat à l'Energie Atomique (CEA), Institut François Jacob, Genoscope, Evry, France. [5] Institute of Experimental Botany of the Czech Academy of Sciences, Centre of the Region Haná for Biotechnological and Agricultural Research, Olomouc, Czech Republic. [6] These authors contributed equally: Caroline Belser, Franc-Christophe Baurens. ✉email: jmaury@genoscope.cns.fr

Long-read technologies are now the standard for generating high-quality assemblies, especially for complex genomes such as plant genomes[1–4]. Although the impact of these technologies is undeniable, they still lack the maturity to reconstruct complete chromosome sequences from telomere to telomere. Generally, assemblies based on long-reads are complemented with long-range data, like optical maps or chromosomal conformation sequencing. Recently, the Telomere-to-Telomere (T2T) consortium proposed a telomere-to-telomere assembly of the X chromosome sequence of the human genome[5]. This high-quality assembly of the human genome was based on a combination of several existing technologies: nanopore sequencing from Oxford Nanopore Technology (ONT), single-molecule real-time (SMRT) sequencing provided by Pacific Biosciences (PACBIO), linked reads sequencing from 10X Genomics (10X) and optical mapping provided by Bionano Genomics (BNG). Even if the final assembly is very contiguous, there are still several gaps, and the complete X chromosome sequence was obtained by manual curation. This huge effort is not possible for all genome projects because it is far too expensive and time-consuming. It is clear that these multilayer assemblies reveal the architecture of complex regions as well as the complete organization of chromosomes, which remained complicated before. Long-range technologies make it possible to organize contigs based on long-reads but they are not able to fill the gaps between these contigs. Indeed, usually complex regions like centromeres or telomeres still contain many gaps, depending on their repetitive content.

We selected the banana genome, a medium-size genome in the plant lineage (~500 Mbp), and hypothesized that recent improvement of the ONT technology, coupled with dedicated DNA extraction protocol and efficient software enables the reconstruction of gapless and Telomere-to-Telomere chromosome sequences.

Banana species are monocotyledonous plants and part of the Zingiberales order and of the Musaceae family. Bananas are mostly cultivated in tropical and subtropical countries, and their fruits are the basis of the diet of several hundred million people and are massively exported to industrialized countries. Four genetic groups have been predicted to be involved in the origins of cultivars, mainly through inter(sub)specific hybridization and with different extents of contribution: *Musa acuminata* including various subspecies (A-genome), *Musa balbisiana* (B-genome), *Musa schizocarpa* (S-genome) and species of the *Australimusa* section (T-genome). Two events appeared during banana domestication: the transition from wild to edible diploids and the emergence of triploids from edible diploids[6–8]. Recent results suggest that edible cultivar origins are more complex than expected, involving multiple hybridization steps, resulting in inter(sub)specific mosaic genomes. They also revealed that additional genetic pools to the ones expected were involved, for which the wild contributors are still unidentified[6]. In addition, large structural variations in form of reciprocal translocations and a few inversions have been characterized in genetic pools involved in cultivar origins and found widespread in cultivated germplasm[6]. The complexity of these genomes underlies the importance of producing high-quality assemblies of banana genomes to decipher their evolutionary history and to support genetic studies.

In this context, two versions of the *Musa acuminata* 'DH-Pahang' genome have already been proposed[9,10]. The first draft version of the genome of this double haploid genotype (V1) was published in 2012 and based on 454, Sanger (fosmids and BAC-ends), and Illumina sequencing. Furthermore, scaffolds were organized using a sparse genetic map, resulting in the anchoring of 63% of the estimated genome size. The second version (V2)

was published in 2016 and added Illumina long-insert sequences, a low-contiguity optical map as well as a more dense genetic map. Martin et al. proposed an assembly of the 11 chromosomes that included 76% of the estimated genome size. Herein we propose to generate a new version (named V4, the third version was an internal assembly not shared with the community) of the DH-Pahang assembly based on nanopore long-reads.

## Results

**Highly contiguous genome assembly of the banana genome.** The efficiency of long-reads sequencing depends on the quality of the DNA extraction. Here, DNA was extracted following a plant-dedicated protocol provided by Oxford Nanopore Technologies ("High molecular weight gDNA extraction from plant leaves"). This protocol was particularly effective and allowed us to obtain long DNA fragments (> 50 kbp). Residual short fragments were filtered out using the Short Read Eliminator (SRE) XL kit (Circulomics, MD, USA). Almost 93 Gb of nanopore sequences were obtained with a single PromethION flowcell R9.4.1. The 5.2 M reads had a N50 of 31.6 kbp and the genome was covered at 17X with reads longer than 75 kbp. This high-quality set of long reads was assembled using several bioinformatics tools and the assembly obtained with NECAT[11] was retained. Indeed the NECAT assembly of this haploid cultivar was the most contiguous (contig N50) and had a larger cumulative size. This assembly was polished first using long reads with Racon[12] and Medaka[13] and then using Illumina short-reads with Hapo-G[14]. This assembly, based on nanopore long reads and without long-range information, was composed of 124 contigs (larger than 50 kbp) and had a cumulative size of 485 Mbp. Half of the assembly size was composed of contigs larger than 32 Mbp and only 16 contigs covered 90% of the total length (Table 1, contig N50 and contig L90). More importantly, the seven largest contigs had a size compatible with complete chromosomes (ranging from 47.7

**Table 1 Comparison of *Musa acuminata* (DH-Pahang) genome assemblies.**

|  | D'hont et al. [9] V1 | Martin et al. [10] V2 | This study V4 |
|---|---|---|---|
| Number of contigs | 29,437 | 19,312 | 124 |
| Cumulative size (bp) | 390,477,446 | 405,522,014 | 484,058,756 |
| N50 (bp) | 28,319 | 43,237 | 32,091,396 |
| L50 | 3,428 | 2,363 | 7 |
| N90 (bp) | 5,108 | 9,026 | 6,704,534 |
| L90 | 16,551 | 10,327 | 16 |
| Longest contig (bp) | 306,660 | 602,020 | 47,719,527 |
| Number of chromosomes | 11 | 11 | 11 |
| Cumulative size (bp) | 331,812,599 | 397,008,016 | 468,821,802 |
| Cumulative size (ACGT only) | 286,824,765 | 363,519,833 | 468,133,046 |
| N50 (bp) | 30,470,408 | 37,593,364 | 43,931,232 |
| L50 | 5 | 5 | 5 |
| N90 (bp) | 25,514,024 | 29,070,452 | 34,826,100 |
| L90 | 10 | 10 | 10 |
| Longest (bp) | 35,439,739 A08 | 44,889,171 A08 | 51,314,288 A08 |
| % of N | 13.56% | 8.44% | 0.68% |
| % of estimated genome | 63.4% | 75.9% | 89.6% |
| % of estimated genome (ACGT only) | 54.8% | 69.5% | 89.5% |
| Complete BUSCO (N = 1,614) | 92.6% | 98.5% | 98.8% |

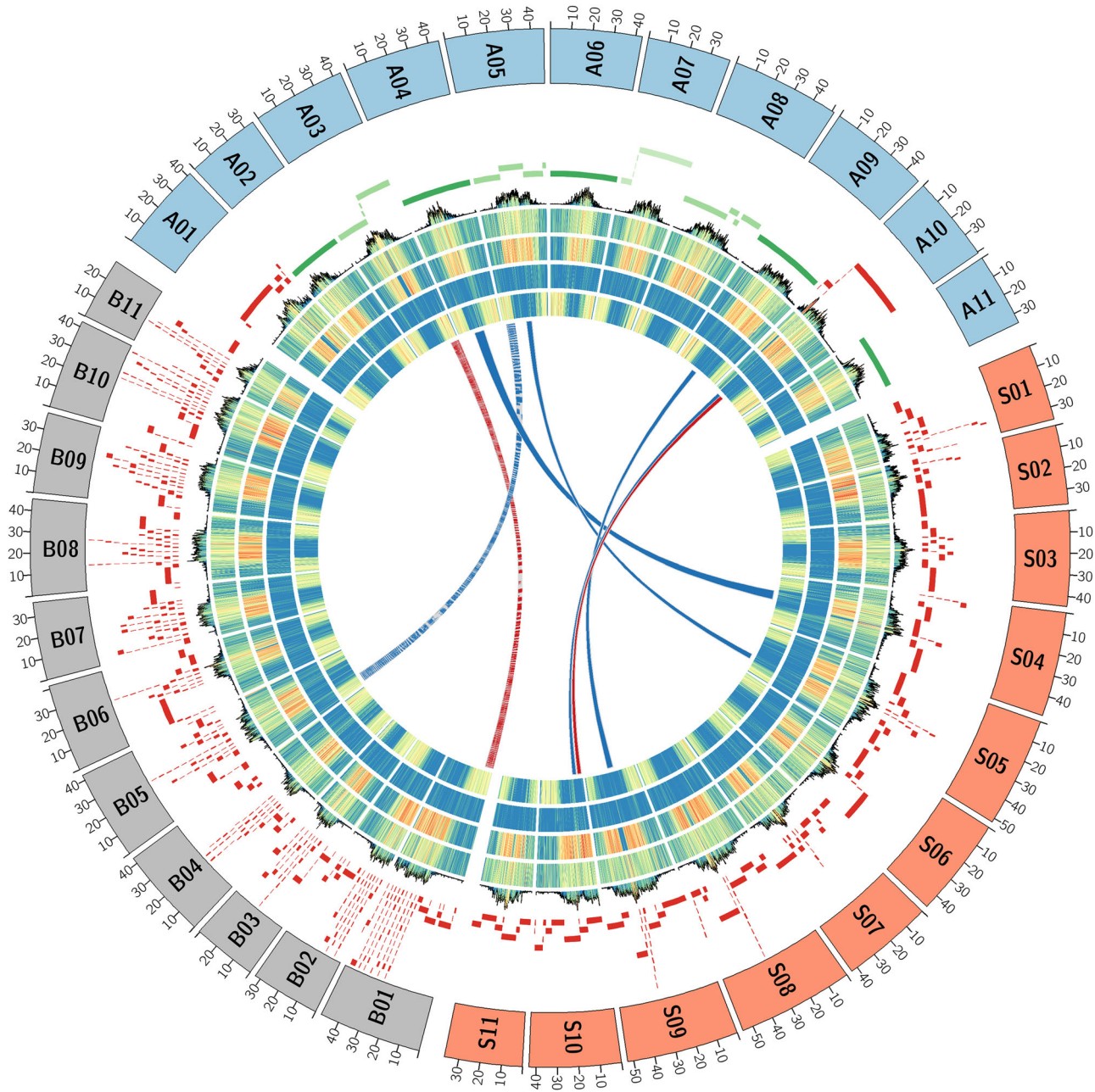

**Fig. 1 Musa genomes architecture comparison.** The tracks represent the following elements (from outer to inner): (1) schematic representation of *M. acuminata* (A), *M. balbisiana* (B) and *M. schizocarpa* (S) chromosome sequences, (2) contigs colored in green if the chromosome sequence is composed of 1–4 contigs, in red if the chromosome sequence is composed of more than 5 contigs. (3) Density of the centromeric repeats. (4) Density of the Gypsy elements. (5) Density of the Copia elements. (6) Density of the DNA transposons. (7) Density of genes. (8) Synteny relationships. The red lines show translocations between B01 and A03 and between S10 and A10. The blue lines show inversions between B05 and A05, S04 and A04, S05 and A05, S09 and A09.

to 32.1 Mbp). The anchoring of contigs was performed following the methodology described in Martin et al. [10]. As expected, the five largest contigs correspond to complete chromosome sequences and harbor telomeric repeats at both extremities (Fig. 1). The six remaining chromosome sequences were composed of a small number of contigs (between four and eight). Interestingly, the remaining gaps are mainly located in rDNA clusters: 5S for chromosomes 1, 3, and 8 and 45S for chromosome 10 or in other tandem and inverted repeats: chromosomes 1 and 5 (Fig. 2a). These rDNA clusters are composed of a large number of tandemly repeated genes and are generally very difficult to assemble. Even if these clusters still contain a few gaps, it is now

possible to decipher the architecture of these large and complex regions. In addition, smaller contigs not anchored to the 11 chromosomes correspond to the chloroplastic and mitochondrial genomes (one and 45 contigs, respectively). A total of 37 contigs were filtered out because they were included in larger contigs and contained highly repeated sequences.

**Validation of telomere-to-telomere chromosome sequences.** A kmer analysis and a first alignment of the largest contigs with the previous version of the DH-Pahang assembly did not reveal chimeric contigs (Supplementary Figs. 1 and 2). In addition, all

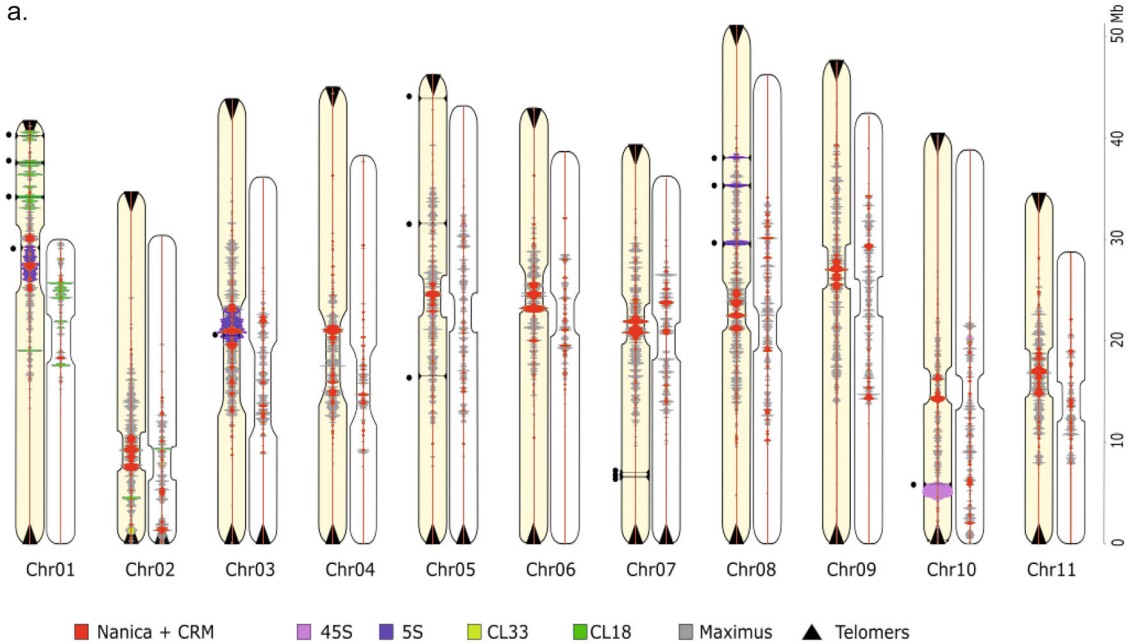

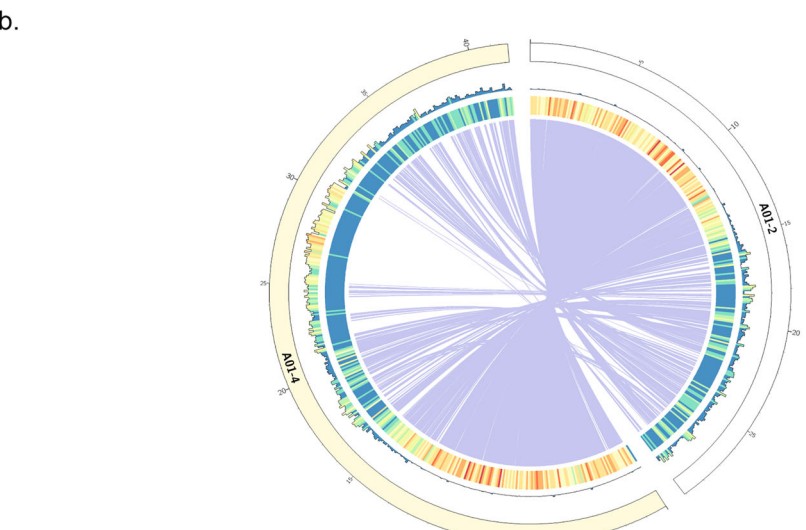

**Fig. 2 Comparison of the V2 and V4 assemblies. a** Localization and density of several repeated elements on chromosome sequences of the V4 (light orange) and V2 (white) assemblies (scale in Mbp on the right) with Nanica LINE and CRM chromovirus Gypsy retrotransposon (red), 5S rDNA (blue), 45S rDNA (violin), tandem repeat cluster CL18 (dark green), tandem repeat cluster CL33 (light green), Maximus Copia retrotransposon (gray) and telomeric sequences (black triangles). Horizontal black lines and black dots correspond to the 15 remaining gaps in the V4 assembly. **b** Comparison of the A01 chromosomes of the V2 and V4 assemblies. Tracks represent the following elements (from outer to inner): (1) density of the centromeric repeats. (2) Density of genes. (3) Synteny relationships between the V2 chromosome 1 and the V4 chromosome 1.

eleven chromosome sequences harbor plant-specific telomeric repeats (T3AG3) at both sides, underlining the complete assembly of chromosome ends.

However, we decided to validate the quality of our assembly using two Bionano optical maps that were generated using the Saphyr instrument commercialized by Bionano Genomics (BNG). High molecular weight DNA was extracted and labeled using two different labeling chemistries independently, the Direct Label chemistry (DLS) and the Nick-Label-Repair and Stain chemistry (NLRS) based on nicking endonucleases. Two optical maps were generated using DLS with the DLE-1 enzyme and NLRS with the BspQI enzyme. The resulting DLE-1 and BspQI optical maps were

469 and 474 Mbp lengths, respectively, and had a N50 of 35 and 16 Mbp, respectively. We used these two optical maps to first validate the contigs, and then order and orient them. As a result, only one contig of 380 kbp, composed of tandem repeated elements, was flagged as conflictual with the optical maps and split into two contigs (Supplementary Fig. 3). All other contigs were in accordance with the maps, which strongly validate the accuracy of the NECAT assembler. The 124 contigs were ordered in 96 scaffolds using the Bionano Solve workflow and the BiscoT[15] software (88 scaffolds correspond each to one contig). In the end, eight of the eleven chromosomes are represented by a single scaffold and the other four remain in two scaffolds. The whole-

genome assembly contains only fifteen gaps that are concentrated in large highly repetitive regions (Supplementary Table 1).

As Illumina paired-end were available for the DH-Pahang genome, an assessment of quality and completeness was performed using Merqury[16] (Supplementary Table 2). As expected the reported completeness is higher for the long-read assembly (95.7% compared to 98.1%). However, the consensus quality (QV) is lower for the nanopore assembly (38.8 vs 49.2). This lower value can be explained by the fact that firstly the nanopore assembly contains regions that are not present in the short-read assembly (such as repetitive regions that are more difficult to polish and can generally contain more errors in nanopore assemblies) and that secondly, the error rate of the nanopore technology is still too high, complicating the correction of the consensus using polishing algorithms. As a control, we calculated the QV score only on the regions shared between both assemblies, and observed a decrease in the difference in quality between the two assemblies (45.9 versus 50.1). This difference may be due to the even higher error rate in the consensus of nanopore assemblies compared to short-read assemblies.

**Comparison of *Musa acuminata* assemblies.** Unsurprisingly, compared to previous versions, the contiguity of our DH-Pahang assembly is greatly improved. The contig N50 goes from a few tens of kbp (28 and 43 kbp for V1 and V2 respectively) to a few tens of Mbp (32 Mbp). More importantly, the cumulative size is closer to the estimated genome size, suggesting that complex regions are better represented in this new release (Table 1). With a very small number of contigs, anchoring on the eleven chromosomes using the genetic map was easier especially in the centromeric regions, which are generally difficult to organize due to their lower density of genetic markers. The size of the DH-Pahang genome was estimated by flow cytometry at 523 Mbp[9]. The 11 chromosome sequences of our long-read assembly cover almost 90% of this estimated size, while the first two versions were largely incomplete (55 and 70% respectively, Table 1). As a consequence, the assembled size of each chromosome sequence has increased, between 7% for chromosome 10−43% for chromosome 1 (Fig. 2a and Supplementary Fig. 2). Chromosome sequences of the two versions were aligned and large genomic regions (>100 kbp) absent in the previous assembly were reported. The 11 chromosome sequences totalized 247 new regions that covered 141.4 Mbp, i.e., 29.2% of the assembly (Supplementary Fig. 4). The largest region, close to 6 Mbp, is localized on chromosome 1 (Fig. 2b). Unsurprisingly, these blocks are mainly composed of repeated elements (more than 85%), and localized in centromeric regions or rDNA clusters (Supplementary Fig. 5). We annotated 246 Mbp of the genome (52.6%) as transposable elements (TE), compared to 152 Mbp in V2, which illustrates the much better representation and completion of these repetitive elements in the V4 assembly (Supplementary Table 3 and Supplementary Fig. 6). Figure 2a shows the distribution of several tandem repeats and TE along the chromosomes, including the Maximus Copia retrotransposons, which are the most abundant TEs in the Pahang genome.

**Architecture of centromeric regions and rDNA clusters.** Earlier cytogenetic analysis showed that a long interspersed element (LINE), named Nanica, is present in the centromeric regions of banana[9,17]. A very few LINE sequences were present in the first release of the assembly despite being present in unassembled reads[9] and they had a scattered distribution on the pericentromeric and centromeric regions of the V2 assembly. In this new assembly, clusters of Nanica tandem repetitions are found grouped in the centromeric regions of all chromosomes (Fig. 2a

and Supplementary Fig. 6). Several elements of chromovirus CRM clade, a lineage of Ty3/Gypsy retrotransposons, were also found restricted to these centromeric regions. Some members of this plant retroelement have been shown to have the ability to target their insertion almost exclusively to the functional centromeres[18,19]. The position of two other tandem repeats (CL18 and CL33) previously identified[17] could also be refined between V2 and V4 and the localization of the main clusters on chromosomes 1 and 2 are in accordance with cytogenetic karyotypes[17]. Regarding the 5S rDNA sequences, in the V2 assembly, they were present in a few numbers in chromosomes 5, 9, and 8 spanning 7,5 kbp of sequences (around 130 gene units) (Supplementary Table 4). In the new assembly, six major loci containing 5S rDNA gene clusters are present accounting for around 7,696 gene units. Three clusters are located on one arm of chromosome 8 representing in total around 2.2 Mbp and two large clusters of 5S rDNA repeat have been integrated to chromosome 1 and 3 centromeric regions, representing around 3.5 and 4 Mbp, respectively. These results are in accordance with previous cytogenetic results[17] and the position of the rDNA clusters have been clarified showing that they colocalize with the centromeric Nanica clusters of these chromosomes. Clusters of 5S rDNA are organized in canonical gene/spacer tandem repeat of different lengths due to the insertion in the spacer of various repeated elements such as Nanica or CRM sequences as observed in the 5S cluster of the centromeric regions of chromosomes 1 and 3 (Fig. 3A, B and c). Furthermore, a large cluster of 1.8 Mbp containing around 110 45S rDNA units, consisting in canonical gene/spacer tandem repeat, is localized on chromosome 10 between positions 4.4 and 6.2 Mbp (Fig. 2a).

**Tandemly duplicated genes (TDGs).** Gene duplication is an important evolutionary mechanism that contributes to the appearance of novel functions and to adaptation. The events leading to gene duplication have contributed to important plant agronomic traits, such as grain quality, fruit shape, and flowering time[20]. A special case of gene duplication relates to genomic/tandem duplication events, which generate, locally, repetitive regions in the genome. These TDGs are generally harder to capture in short-read assemblies, especially in the case of recent multi-copy clusters. We found 1,700 genes that have been annotated only in our long-read assembly. These genes are distributed over the different chromosomes, with chromosomes 1 and 10 having the greatest number of new genes, 13.3 and 14.3% respectively (Supplementary Table 5). Interestingly, a large proportion (38.3%) of these genes are TDGs included in a gene cluster and the proportion of TDGs in new genes is higher when compared to the whole gene catalog (38.3% versus 9.9%).

By focusing on TDGs and detecting gene clusters in the short and long-read assemblies, we found 31% more clusters in the V4 compared to V2 assembly (1,134 compared to 866 clusters). These blocks of TDGs contain respectively 3,649 and 1,134 genes. The largest in the long-read assembly contains 38 genes on chromosome 7 (between 31.8 and 32.8 Mbp) and was split into two smaller clusters of 11 and 9 genes in the V2. This TDG cluster is located in a region with several gaps in previous versions, and we found three regions (165, 111 and 110 kbp) between positions 31.8 and 32.3 Mbp of the chromosome 7 that are specific to the V4 assembly. These new regions allow the creation of a complete cluster of TDGs (Fig. 4a) which contain motifs of the terpene synthase family that are responsible for the synthesis of terpenoid compounds playing a role in plant flavor[21] and more generally in the interactions between the plant and its environment[22,23]. This family is known to contain TDGs and is expanded in several plant species[24].

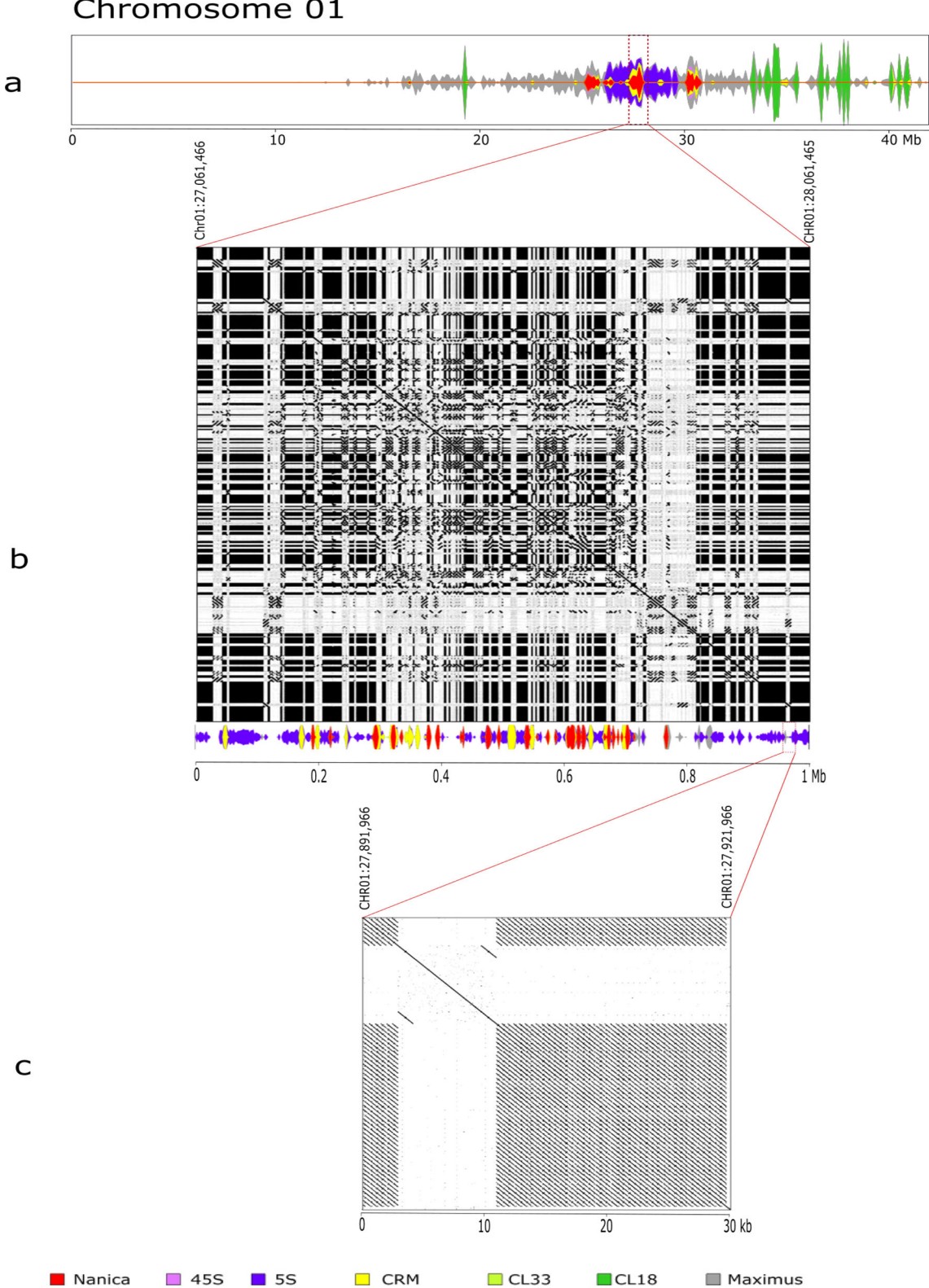

**Fig. 3 Fine structure and density of main (peri)centromeric repeated sequences on chromosome 1.** Nanica LINE (red), CRM chromovirus Gypsy retrotransposon (yellow), 5S rDNA (lilac), tandem repeat cluster CL18 (dark green), Maximus Copia retrotransposon (gray) are represented on: (**a**) the entire chromosome 1: (**b**) a zoom and a dot-plot alignment of a 1 Mbp segment in the centromeric region containing Nanica, 5S rDNA, and CMR repeats, (**c**) a zoom and a dot-plot alignment of a 30 kbp segment containing 5S rDNA repeats and a CMR.

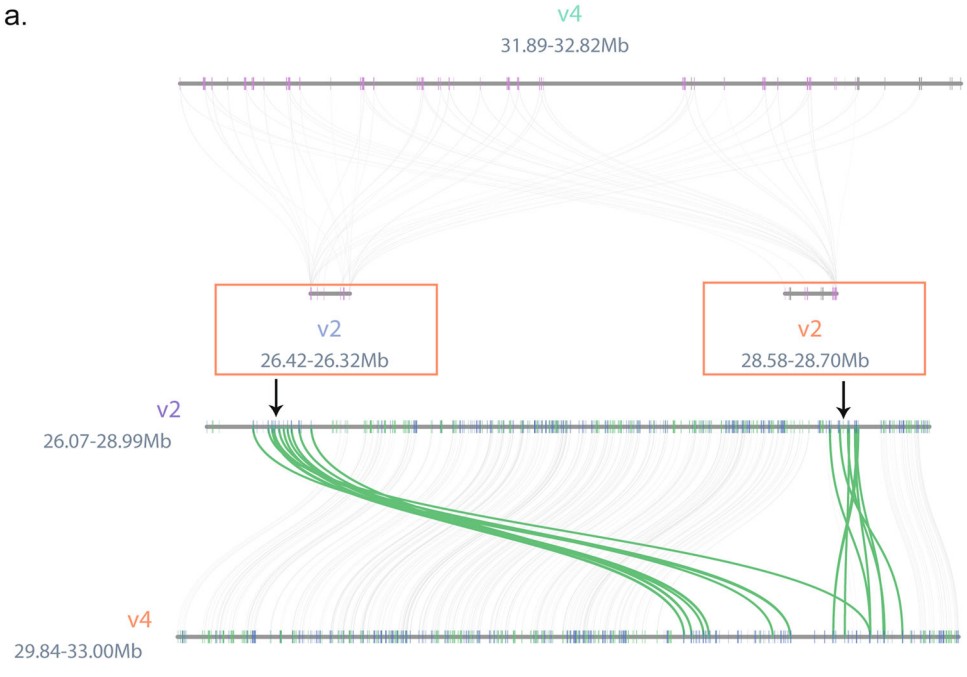

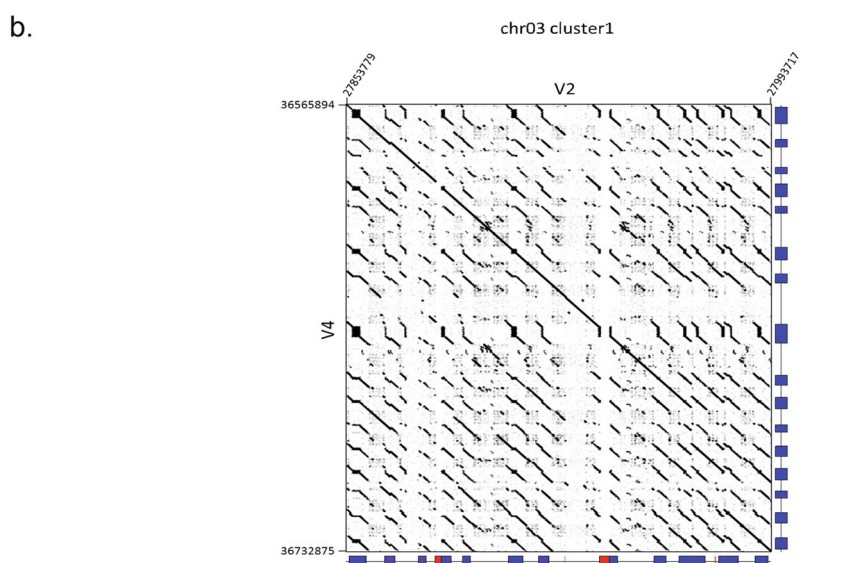

**Fig. 4 Comparison of tandemly duplicated gene regions in the V2 and V4 assemblies. a** Synteny visualization of gene clusters of the terpene synthase family. The cluster is located between 31.89 and 32.82 Mbp on the V4 (chromosome 7). Gene synteny relationships are colored in green, and genes are colored according to their orientation (blue if forward and green otherwise). **b** Comparison of the structure of a NLR cluster on chromosome 3. The predicted NLR loci for each version are represented by blue boxes on the *x*- and *y*-axis of the dot plots. Red boxes represent regions bearing undetermined nucleotides. Region coordinates are also indicated.

**Resistance genes**. Plant disease resistance genes encoding proteins with nucleotide-binding leucine-rich repeat (NLR) domains are often clustered in genomes, sometimes forming large, rapidly evolving clusters of highly homologous genes[25]. The NLR-annotator program[26] allows the identification of NLR loci i.e., genomic regions likely associated with an NLR gene (or pseudogene). A total of 128 NLR loci were detected in this assembly compared to 111 loci in the V2 assembly (Table 2, and Supplementary Data 1 and 2). Four major clusters of NLR loci were found in this assembly: two in chromosome 3, one in chromosome 7, one in chromosome 10 (Fig. 4b and Supplementary

Fig. 8). They all have a larger size compared to V2, with sizes ranging from 132 up to 227 kbp and additional detected NLR loci. These clusters were improved in sequence quality with a complete absence of undetermined nucleotides within the corresponding genomic regions (Supplementary Table 6).

**Comparison of A, B and S-genome assemblies**. The B and S genomes have already been recently sequenced using a long-read strategy[27,28]. Taking into account this new version of the *M. acuminata* genome, three high-quality banana genomes are now available. The A and S genomes were sequenced using ONT while

| Table 2 Comparison of gene prediction statistics. | | |
|---|---|---|
| Reference | Martin et al. [10] V2 | This study V4 |
| # Number of genes | 35,276 | 36,979 |
| #Exons per spliced gene (avg:med) | 6.08: 5 | 6.05: 4 |
| Gene sizes (avg:med) | 4,542: 2,824 | 4,604.68: 2,758 |
| CDS sizes (avg:med) | 1,171: 981 | 1,180: 972 |
| Complete BUSCO (N = 1,614) | 98.5% | 98.8% |
| NLR loci | 111 | 128 |

| Table 3 Comparison of A (*Musa acuminata*), B (*Musa balbisiana*) and S (*Musa schizocarpa*) genomes assemblies. | | | |
|---|---|---|---|
| | *Musa acuminata* | *Musa balbisiana* | *Musa schizocarpa* |
| Number of contigs | 124 | 3,787 | 379 |
| Cumulative size | 484,058,756 | 491,421,783 | 517,486,196 |
| N50 (bp) | 32,091,396 | 1,801,976 | 6,493,909 |
| L50 | 7 | 59 | 24 |
| N90 (bp) | 6,704,534 | 56,360 | 1,047,001 |
| L90 | 16 | 578 | 84 |
| Longest contig (bp) | 47,719,527 | 14,987,599 | 18,138,554 |
| Number of chromosomes | 11 | 11 | 11 |
| Cumulative size | 468,821,802 | 430,021,147 | 496,921,565 |
| Cumulative size (ACGT only) | 468,133,046 | 429,290,714 | 490,105,212 |
| % Anchored sequences | 96.8% | 87.3% | 94.7% |
| N50 (bp) | 43,931,232 | 42,323,520 | 46,993,692 |
| L50 | 5 | 5 | 5 |
| N90 (bp) | 34,826,100 | 30,518,812 | 36,762,080 |
| L90 | 10 | 10 | 10 |
| Longest (bp) | 51,314,288 | 48,736,620 | 54,858,060 |
| Number of gaps | 15 | 683 | 166 |
| Estimated genome size | 523 Mb | 520 Mb | 587 Mb |
| % of estimated genome size | 89.6% | 82.6% | 84.6% |
| Number of annotated genes | 36,979 | 35,148 | 32,809 |
| Complete BUSCO (N = 1,614) | 98.8% | 96.9% | 97.6% |

the B genome was sequenced using the PACBIO technology. Interestingly, the two ONT assemblies have a higher contiguity (contig N50 of 32 and 6.5 Mbp compared to 1.8 Mbp) suggesting the usage of longer reads, or the difficulty to extract and sequence long DNA fragments with the PACBIO device (Table 3). Indeed, the PACBIO library was size-selected in order to obtain fragments around 20 kbp[28], which was perhaps an optimal condition for PACBIO sequencing at this time. The B genome was assembled from reads with an N50 of 16.6 kbp whereas A and S genomes were assembled with reads having an N50 of 31.6 and 24.4 kbp respectively. As a consequence, in addition, we noticed that chromosome sequences of the A and S genomes contain fewer gaps. A difference between the PACBIO and ONT sequencing technologies is already mentioned[29]. The eleven A and S chromosome sequences contain 15 and 166 gaps respectively whereas B chromosome sequences contain 683 gaps and no chromosome sequence is gapless (Fig. 1). Centromeric regions, detected with centromeric repeats, are very fractionated in the case of the PACBIO-based assembly (from 24 contigs for the chromosome 7 to 111 contigs for the chromosome 1), underlying the importance of ultra-long reads to resolve these highly repetitive regions. However, sequencing technologies evolve rapidly and this comparison does not reflect the current capability of each technology.

Overall, the synteny conservation between the three genomes is high, we detected one inversion between the chromosome B05 of *Musa balbisiana* and the chromosome A05 of *Musa acuminata* and a translocation between the chromosome B01 and the chromosome A03 as already reported[28] (Fig. 1 and Supplementary Fig. 9). Four inversions between *Musa schizocarpa* and *Musa acuminata* were also detected on the chromosomes S10, S04, S05, and S09 and one translocation on chromosome A10 (Fig. 1 and Supplementary Fig. 10). The corresponding regions on chromosome A10 contain the 45S rDNAS gene cluster. The contigs organization in these five regions was manually validated in the two genome assemblies using optical maps.

## Discussion
Long read sequencing technology emergence has paved the way for high-quality genome assemblies. The rapid evolution of the DNA extraction protocols now allows the community to sequence very long DNA sequences. Coupled with the evolution of the bioinformatic tools, the generation of high-quality assemblies has been greatly simplified. The latest improvements of the ONT technology, especially the base-calling efficiency, result in a decrease of the error rate. Several assembly tools were specially developed around long reads and are able to manage noisy reads and have their specificity, as well as a specific margin of progress. We think that it is still important to use the latest release of several assemblers and choose the most efficient for each genome assembly project.

In this study, we combined recent development from DNA extraction, sequencing, and genome assembly and showed that plant chromosome sequences can now be assembled in a single contig, gapless, and from telomere to telomere, at least to a certain extent. We chose the genome of *Musa acuminata*, the first monocotyledonous species sequenced outside Poales, because its reference genome, even of low-quality, has been widely used and constitutes an important resource for the scientific community. To date, three *Musa* species: *Musa acuminata*, *Musa balbisiana,* and *Musa schizocarpa* are available at the chromosome-scale. These three species are of particular interest because they are involved in the origin of banana cultivars; their high-quality genome assemblies will thus be a valuable resource to explore the evolutionary history and biology of current banana cultivars. We reported that the PACBIO assembly of *M. balbisiana* is more fragmented which can be related to the input read size and underline the importance of long reads to resolve highly repetitive regions.

We generated a highly contiguous assembly of the eleven chromosome sequences of *Musa acuminata* of which five were obtained in a single contig. At the same time, optical maps were used to validate the nanopore assembly. It is important to mention that only one small contig, essentially composed of repetitive elements, was detected as a potential chimera underlining the high quality of the contigs produced by the NECAT assembler. However, it should be noted that homozygous material was used and this limited complexity may not be representative of the situation in other species. All eleven chromosome sequences, build with the help of a genetic map, contain telomeric repeats at both ends, which is an important element in asserting on the one hand that the reconstruction of the chromosome sequences is of good quality and on the other hand that the still missing part of the genome is contained in the remaining fifteen gaps, although 7 are of unknown length. One of the advantages of optical maps is that the size of the gaps can be estimated if the map is sufficiently contiguous (Supplementary Table 1 and Supplementary Fig. 11).

Comparison of the distribution of repeated sequences (tandem repeat and TE) between V2 and V4 showed that the integration of these elements that were typically difficult to assemble with past technologies are greatly improved in the new assembly and are now very congruent with cytogenetic karyotype. All centromeres are now clearly identified with large clusters of Nanica LINE tandem repeats and CMR TE, and in addition, for two of them large clusters of 5S rDNA tandem repeats. Such a case of recruitment of 5S rDNA gene array in centromere was also reported in one of the switchgrass chromosomes[30]. This high-resolution of centromeric regions opens new avenues to study how satellites repeats originate and evolve in the centromeric region and more generally to better understand the organization and functioning of centromeres that are essential chromosomal domains for kinetochore assembly and correct chromosome segregation[31,32]. In addition, chromosome reciprocal translocations were recently shown to have accompanied subspecies evolution in *Musa*[6], and some of them have their breakpoints in centromeric regions. Having access to the sequence of these centromeric regions will permit investigating the mechanism and sequences involved in the origin of these translocations. Finally, comparison with the *Musa acuminata* V2 assembly highlights a higher proportion of each class of transposable elements, and a large amount of additional sequences in the centromeric regions, like Nanica elements, or large retro-transposon derivatives.

It is often mistakenly thought that short-read assemblies are complete at the gene level. This hypothesis is mainly based on the results given by the BUSCO software, which only focuses on single-copy genes. Accordingly, the gene content completeness was already high, in previous *M. acuminata* assembly versions, according to the BUSCO score. However, here, using ultralong reads, we were able to assemble many additional copies of TDG clusters which contain important gene families like terpene synthases or disease resistance genes. Banana crops are currently particularly threatened by diseases including Black leaf streak disease that requires massive use of pesticide[33] and by a new strain of Fusarium wilt (Tropical Race 4) that is currently spreading around the world and for which no chemical control is possible[34]. This new assembly will facilitate the search for resistance genes to these devastating diseases[35].

Finally, we showed that gapless and telomere-to-telomere assembly of chromosome sequences is now possible thanks to long-read sequencing, at least in the case of homozygous genomes. The critical point remains the DNA extraction protocols that generally need adaptation for each species. These closed assemblies will allow new discoveries and will shed new light on these genomes in particular in complex repetitive regions such as centromeres, which have essential biological function but are so far poorly characterized.

## Methods

**Plant material**. Double haploid *Musa acuminata* spp *malaccensis* (*DH-Pahang*) plant material was obtained from the CRB Plantes Tropicales Antilles CIRAD-INRA Guadeloupe under the collection number PT-BA-00461.

**DNA extraction**. For Illumina sequencing libraries, DNA was extracted using a modified mixed alkyl trimethyl ammonium bromide (MATAB) procedure[36]. A total of 2 g freshly harvested leaves was ground in liquid nitrogen with a mortar and pestle and immediately transferred to 12 ml of 74 °C prewarmed extraction buffer containing 100 mM Tris-HCl, pH 8, 20 mM EDTA, 1.4 M NaCl, 2% w/v MATAB, 1% w/v PEG6000 (polyethylene glycol), 0.5% w/v sodium sulfite and 20 mgl$^{-1}$ RNAse A. Crude extracts were maintained for 20 min at 74 °C, extracted with an equal volume of chloroform-isoamyl alcohol (24:1) and transferred to clean tubes. DNA was recovered by centrifugation after adding 10 ml isopropanol. DNA precipitates were briefly dried, washed with 2 ml of 70% ethanol and resuspended in 1 ml sterile water. Extract quality was evaluated using pulse-field gel electrophoresis for size estimation and spectrophotometry (A260/A280 and A260/A230

ratios) for purity estimation. DNA samples with a fragment size above 50 kbp, a A260/A280 ratio close to 2 and a A260/A230 ratio above 1.5 were kept.

In order to generate long reads on the Oxford Nanopore Technologies devices, high-quality and high-molecular-weight DNA is needed. To that end, DNA was isolated following the protocol provided by Oxford Nanopore Technologies, "High molecular weight gDNA extraction from plant leaves" downloaded from the ONT Community in March, 2019 (CTAB-Genomic-tip). This protocol involves a conventional CTAB extraction followed by purification using the commercial Qiagen Genomic tip (QIAGEN, MD, USA), but size selection was performed using Short Read Eliminator XL (Circulomics, MD, USA) instead of AMPure XP beads. Briefly, 1.2 g of leaves were cryoground in liquid nitrogen. The fine powder was transferred to 20 mL of Carlson buffer (100 mM Tris-HCl pH 9.5, 2% CTAB, 1.4 M NaCl, 1% PEG 8000, 20 mM EDTA, 0.25% b-mercaptoethanol (v/v)) prewarmed to 65 °C. Then 40 μl of RNase A (100 mg/ml) was added before incubation at 65 °C for 1 h (with intermittent agitation). Proteins removal was performed by addition of one volume of chloroform and centrifugation at 5,500×g for 10 min at 4 °C. DNA was then precipitated with 0.7 V of isopropanol and centrifugation at 5,500×g for 30 min at 4 °C. The pellet was then purified using the Qiagen Genomic-tip 100/G, following the manufacturer's instruction: DNA pellet was first dissolved at 50 °C for 15 min in 9.5 mL of G2 buffer before loading onto the pre-equilibrated Genomic-tip column. Purified gDNA was finally precipitated with 0.7 volumes of isopropanol, washed with 2 ml of 70% ethanol, dried, and eluted in 100 μL of TE Buffer. DNA was quantified by a dsDNA-specific fluorometric quantitation method using Qubit dsDNA HS Assays (ThermoFisher Scientific, Waltham, MA). DNA quality was checked on a 2200 TapeStation automated electrophoresis system (Agilent, CA, USA) (Supplementary Fig. 12).

Generating optical maps requires high molecular weight (HMW) DNA. Here HMW DNA of *M. acuminata* DH Pahang was prepared according to Safář et al. [37] with several modifications. Briefly, 0.5 cm long segments of leaf midribs and young leaf tissues were fixed for 20 min at 4 °C in Tris buffer (10 mM Tris, 10 mM EDTA, 100 mM NaCl, pH 7.5) containing 2% formaldehyde. After three 5 min washes in Tris buffer, the segments were homogenized using chopping by a razor blade in petri dish containing 1 ml of ice-cold IB buffer (15 mM Tris, 10 mM EDTA, 130 mM KCl, 20 mM NaCl, 1 mM spermine, 1 mM spermidine and 0.1% Triton X-100, pH 9.4) and immediately before use, 33 μl of β-mercaptoethanol were added to 10 ml of IB buffer. Nuclei suspension was passed through a 50 μm nylon mesh and stained with DAPI at a final concentration of 2 μg/mL. Six batches of 900,000 G1-phase nuclei were sorted into 77 μl of IB buffer with β-mercaptoethanol in 1.5 ml polystyrene tubes using a FACSAria SORP flow cytometer and sorter (Becton Dickinson, San José, CA, United States) equipped with solid-state UV laser. One 20 μl agarose mini-plug was prepared from each batch of nuclei according to Šimková et al. [38]. Miniplugs were washed and solubilized using agarase enzyme (Thermo Fisher Scientific) to release high molecular weight (HMW) DNA. HMW DNA was further purified by drop dialysis and was then homogenized a few days prior to the quality control.

The concentration and purity of the extracted DNA were evaluated using a Qubit fluorometer (Thermo Fisher Scientific) and a Nanodrop spectrophotometer (Thermo Fisher Scientific). DNA integrity was checked by pulsed-field gel electrophoresis (Pippin Pulse, Sage Science). DNA molecules were detectable between 50 and 300 kbp in size.

**Illumina PCR-free library preparation and sequencing**. DNA (1.5 μg) was sonicated to a 100−1500 bp size range using a Covaris E220 sonicator (Covaris, Woburn, MA, USA). The fragments were end-repaired and 3′-adenylated. Illumina adapters were added using the Kapa Hyper Prep Kit (KapaBiosystems, Wilmington, MA, USA). The ligation products were purified with AMPure XP beads (Beckman Coulter Genomics, Danvers, MA, USA). The libraries were quantified by qPCR using the KAPA Library Quantification Kit for Illumina Libraries (Kapa-Biosystems), and the library profiles were assessed using a DNA High Sensitivity LabChip kit on an Agilent Bioanalyzer (Agilent Technologies, Santa Clara, CA, USA). The libraries were sequenced on an Illumina HiSeq2500 instrument (Illumina, San Diego, CA, USA) using 250 base-length read chemistry in paired-end mode.

After the Illumina sequencing, an in-house quality control process was applied to the reads that passed the Illumina quality filters. The first step discards low-quality nucleotides ($Q < 20$) from both ends of the reads. Next, Illumina sequencing adapters and primer sequences were removed from the reads. Then, reads shorter than 30 nucleotides after trimming were discarded. These trimming and removal steps were achieved using in-house-designed software based on the FastX package[39]. The last step identifies and discards read pairs that are mapped to the phage phiX genome, using SOAP aligner[40] and the Enterobacteria phage PhiX174 reference sequence (GenBank: NC_001422.1). This processing, described in Alberti et al. [41], resulted in high-quality data.

**PromethION library preparation and sequencing**. The library was prepared according to the following protocol, using the Oxford Nanopore SQK-LSK109 kit. Genomic DNA fragments (4 μg) were repaired and 3′-adenylated with the NEB-Next FFPE DNA Repair Mix and the NEBNext® Ultra™ II End Repair/dA-Tailing Module (New England Biolabs, Ipswich, MA, USA). Sequencing adapters provided by Oxford Nanopore Technologies (Oxford Nanopore Technologies Ltd, Oxford,

UK) were then ligated using the NEBNext Quick Ligation Module (NEB). After purification with AMPure XP beads (Beckmann Coulter, Brea, CA, USA), half of the library was mixed with the sequencing buffer (ONT) and the loading bead (ONT) and loaded on a PromethION R9.4.1 flow cell. The second half of the library was loaded on the flow cell after a Nuclease Flush using the Flow Cell Wash Kit EXP-WSH003 (ONT) according to the Oxford Nanopore protocol. Reads were basecalled using Guppy version 4.0.1. The nanopore long reads were not cleaned and raw reads were used for genome assembly.

**Optical mapping**. The Direct Label and Stain (DLS) labeling (using the DLE-1 enzyme) and the Nick Label Repair and Stain (NLRS) labeling (using the BspQI enzyme) protocols were performed according to Bionano Genomics with 750 and 600 ng of DNA respectively. The Chip loadings were performed as recommended by Bionano Genomics.

**Long reads-based genome assembly**. We generated three samples of reads: all reads, 30X of the longest reads, and 30X of the filtlong[42] highest-score reads. We then applied four different assemblers, Smartdenovo[43], Redbean[44], Flye[45], and NECAT[11] on these three subsets of reads (Supplementary Table 7), with the exception of NECAT being only launched with all reads, as it applies a down-sampling algorithm in its pipeline. Smartdenovo was launched with -k 17, as advised by the developers in case of larger genomes and -c 1 to generate a consensus sequence. Redbean was launched with '-xont -X5000 -g450m' and Flye with '-g 450m'. NECAT was launched with a genome size of 450 Mbp and other parameters were left as default. After the assembly phase, we selected the best assembly (NECAT with all reads) based on the cumulative size and contiguity. The assembler output was polished one time using Racon[12] with Nanopore reads, then one time with Medaka[13] and Nanopore reads, and two times with Hapo-G[14] and Illumina PCR-free reads (Supplementary Table 8).

**Assembly validation**. The DLE-1 map was generated using the Direct Label and Stain (DLS) technology and the BspQI map using the Nick Label Repair and Stain (NLRS) technology. Genome map assemblies were performed using Bionano Solve Pipeline version 3.3 and Bionano Access version 1.3.0. We used the parameter "Add Pre-Assembly" which produced a rough assembly. This first result was used as a reference for a second assembly, using the parameters "non-haplotype without extend and split". We filtered out molecules smaller than 150 kbp and molecules with less than nine labeling sites (Supplementary Tables 9 and 10). The nanopore contigs were then validated using the two Bionano maps and organized with the scaffolding procedure provided by Bionano Genomics (Supplementary Fig. 13). Negative gap sizes were checked and corrected using the BiscoT software[15] to avoid artifactual genomic duplications (Supplementary Table 1). As recommended by the BiscoT authors, we performed a last iteration of Hapo-G[14] to polish merged regions (Supplementary Table 11). In order to obtain a quality score used to compare the different versions of the assembly, we downloaded and used merqury[16] version 1.3 (git commit 6b5405e). We first used the included best_k.sh script with a tolerable collision rate of 0.0001 and a genome size of 500 Mbp, which gave us an estimated best k-mer size of 21. Then, we used meryl[46] version 1.3 (git commit 3400615) to compute the reads k-mer counts via the meryl count command with default parameters. Merqury was then launched on two sets of sequences. The first one consists of the V1 and V4 assemblies, in order to compare them in their globality. As the V4 assembly is larger than V1, we aligned the V1 assembly to the V4 assembly using minimap2 and kept alignments that were larger than 50 kbp. Regions of both assemblies corresponding to these alignments were extracted and used as a second set that we used as input to merqury, to compare assemblies only in regions that are shared.

**Chromosome sequences reconstruction**. DH-Pahang sequences were anchored on chromosomes using segregating markers obtained from the selfing of the 'Pahang' accession PT-BA-00267, described in Martin et al. [10] (Supplementary Table 12 and Supplementary Fig. 14). Data are available on the Banana Genome Hub[47] in the download section under 'AF-Pahang marker matrix file' and 'AF-Pahang marker sequence (FASTA)' for coded segregating markers and marker sequence respectively. Sequences anchoring was performed following methodology described in Martin et al. [10]. The complete process was performed using scaff-hunter tools[48] available at the South Green platform.

In addition, based on scaffold BLAST against *Musa acuminata* chloroplast sequence[49], *Musa acuminata* putative 12 mitochondrial scaffolds[10] and *Phoenix dactylifera* protein sequences[50], 1 putative chloroplastic (corresponding to one initial contig), and 45 putative mitochondrial scaffolds (corresponding to 45 initial contigs) were identified in the assembly. The 45 putative mitochondrial scaffolds were grouped into a sequence named *putative_mito_scaff*. The chloroplastic scaffold was discarded from the assembly as the chloroplast genome of DH-Pahang was already fully assembled and published[49].

The 37 remaining scaffolds (cumulative size of 5.2 Mbp) (corresponding each to one initial contig) showed a strong BLAST homology to larger scaffolds included in the chromosome sequences (36 with more than 95% of their length and 1 with more than 88% of its length). Investigation (dot-plot analysis using gepard v1.30[51] and BLAST against nr/nt of ncbi) of these scaffolds revealed a repetitive nature,

most of them corresponding to rDNA sequences. Because of their strong homology to scaffolds included in the chromosome sequences these scaffolds were discarded from the assembly.

**Gene prediction**. Repeats in the genome assembly were masked using Tandem Repeat Finder[52] for tandem repeats and RepeatMasker[53] for simple repeats, as well as known repeats included in RepBase[54]. In addition, known *Musa* transposable elements (from D'Hont et al. [9]), were detected using RepeatMasker.

Gene prediction was done using proteomes from homologous species, *Musa acuminata* (UP000012960), *Oryza. sativa* (UP000059680), *Phoenix dactylifera* (UP000228380), *Musa schizocarpa* (www.genoscope.cns.fr/plants) and *Musa balbisiana* (banana-genome-hub.southgreen.fr).

The proteomes were aligned against the genome assembly in two steps. Firstly, BLAT[55] (default parameters) was used to quickly localize corresponding putative genes of the proteins on the genome. The best match and matches with a score ≥ 90% of the best match score were retained. Secondly, the alignments were refined using Genewise[56] (default parameters), which is more precise for intron/exon boundary detection. Alignments were kept if more than 80% of the length of the protein was aligned to the genome.

To allow the detection of UTRs in the gene prediction step, we aligned not only the protein of *M. acuminata*, but also the virtual mRNAs of the *M. acuminata* predicted genes[57] on a masked version of the genome assembly. Then, transcript sequences of *M. acuminata* predicted genes were aligned by BLAT (default parameters) on the masked genome assembly. Only the alignments with an identity percent greater or equal to 90% were kept. For each transcript, the best match was selected based on the alignment score. Finally, alignments were recomputed in the previously identified genomic regions by Est2Genome[58] in order to define precisely intron boundaries. Alignments were kept if more than 80% of the length of the transcript was aligned to the genome with a minimal identity percent of 95%.

To proceed to the gene prediction, we integrated the protein homologies and transcript mapping using a combiner called Gmove[59]. This tool can find CDSs based on genome located evidence without any calibration step. Briefly, putative exons and introns, extracted from the alignments, were used to build a simplified graph by removing redundancies. Then, Gmove extracted all paths from the graph and searched for open reading frames (ORFs) consistent with the protein evidence. A selection step was applied to all candidate genes, essentially based on gene structure. Also, all gene predictions included in a genomic region tagged as transposable elements were removed from the gene set (Table 2). The completeness of the predicted genes was assessed with BUSCO[60] version 5 (embryophyta dataset odb10).

The search for NLR loci was performed using NLR-annotator tools[26] that scan specifically the 6 reading frames of the nucleotide sequence for the presence of 19 NLR-associated motifs and reconstruct a potential NLR locus which might correspond to a complete or partial gene and might also be a pseudogene.

**Transposable element detection**. Transposable elements were detected using RepeatMasker[53] associated with the TE Musa library[9] and CR sequences[18]. The same procedure was used to detect TEs in the *Musa acuminata* V2, V4, *Musa balbisiana* and *Musa schizocarpa* assemblies. The gff output file was converted into a bed file and the TE coverage was calculated using bedtools[61] coverage (version v2.29.2-17-ga9dc5335) on a 100 kbp window. Centromeric boundaries were defined using the density of daterra-Maximus, ITS-5S, ITS-18S, ITS-26S, Nanica, maca-Angela, caturra-Reina elements. TEs were grouped following Wicker et al. classification[62].

**Genome assemblies comparisons**. The synteny relationships between *Musa balbisiana*, *Musa schizocarpa* and *Musa acuminata* V4 were determined using Assemblytics[63]. First, genome sequences were aligned against each other using nucmer[64] version 3.23 (-maxmatch -l 100 -c 500) as recommended by the Assemblytics authors. Assemblytics was launched on the nucmer delta file (unique_length_required = 10000). Figures 1 and 2b were generated using the Circos software[65]. In the same way, each chromosome sequence of the *Musa acuminata* V4 was aligned against its relative chromosome sequence of *Musa acuminata* V2 using nucmer version 3.23 (-r −1 -l 10000) and dot plots were generated using the mummerplot command.

**Detection of specific regions of the V4 assembly**. New regions of the *Musa acuminata* V4 were determined using blast[66] (ncbi-tools/6.1.20120620) alignment between the chromosomes of each assembly version. Regions of the V4 assembly larger than 100 kbp without any alignment to the V2 assembly were considered new. In addition, the *Musa acuminata* V2 gene predictions were aligned (see Gene prediction section) on the V4 assembly, and the positions of the genes were compared with the V4 gene catalog using bedtools[61] (version bedtools-2.29.2) (-v option). Genes from the V4 assembly without any correspondence in the V2 were considered new.

**Detection of tandemly duplicated genes**. An all-against-all comparison of the *Musa acuminata* V4 proteins was performed using Diamond[67] (version 0.9.24). Mapping output was filtered according to the following parameters: an e-value

lower than 10e−20 and a coverage of the smallest protein greater than 80%. Genes were considered as tandemly duplicated if they were co-localized on the same chromosome and not distant from more than 10 genes to each other. Figure 4a was realized using the MCscan tool[68] with the two following commands: jcvi.-compara.synteny (--iter=1) and jcvi.graphics.synteny (--glyphcolor = orientation).

**Statistics and reproducibility**. No statistical tests were used in this study, and to allow the reproducibility of our analysis and results, all the sequencing data are available in public databases and the scripts developed to generate the figures are available on Zenodo and on a Github repository, as described in the data and code availability sections.

**Reporting summary**. Further information on research design is available in the Nature Research Reporting Summary linked to this article.

## Data availability
All the supporting data are included in three additional files which contain (a) Supplementary Tables 1−12 and Supplementary Figs. 1−14, (b) Supplementary Data 1 (position of NLR genes in the V4 assembly) and (c) Supplementary Data 2 (position of NLR genes in the V2 assembly). The genome assembly is freely available at http://www.genoscope.cns.fr/plants and http://banana-genome-hub.southgreen.fr. The ONT, Illumina, and Bionano Genomics data are available in the European Nucleotide Archive under the following projects PRJEB35002.

## Code availability
All the code and data used to generate the figures are available on Zenodo[69] and on a Github repository https://github.com/institut-de-genomique/Pahang-associated-data

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

## Acknowledgements

This work was supported by the Genoscope, the Commissariat à l'Energie Atomique et aux Energies Alternatives (CEA), France Génomique (ANR-10-INBS-09-08), the Center de coopération Internationale en Recherche Agronomique pour le Développement (CIRAD) and Agropolis Fondation (ID 1504-006) 'GenomeHarvest' project through the French Investissements d'avenir program (Labex Agro: ANR- 10-LABX-0001-01). EH and JD were supported by ERDF project 'Plants as a tool for sustainable global development' (No. CZ.02.1.01/0.0/0.0/16_019/0000827). The authors thank the staff of Oxford Nanopore Technology Ltd for technical help, Jitka Weiserová, Eva Jahnová and Dr. Jan Vrána for their help with the material preparation, the CRB Plantes Tropicales Antilles CIRAD-INRA Guadeloupe France for providing the plant materials and the CIRAD – UMR AGAP HPC Data Center of the South Green Bioinformatics platform (http://www.southgreen.fr) for providing computational resources.

## Author contributions

K.L. extracted the sequenced DNA. E.H., and J.D. prepared HMW DNA for optical mapping. K.L. extracted the plugs. C.C. realized the bionano experiments. K.L., C.C., and A.L. optimized and performed the sequencing. C.B., B.I., B.N., N.Y., F.C.B., G.M., and J.M.A. performed the bioinformatic analyses. A.D. and J.M.A. conceived the project. C.B., B.I., B.N., K.L., C.C., E.H., G.M., A.D., and J.M.A. wrote the article. A.D., P.W., and J.M.A. supervised the study.

## Competing interests

The authors declare the following competing interests, J.M.A. received travel and accommodation expenses to speak at Oxford Nanopore Technologies conferences. J.M.A. and C.B. received accommodation expenses to speak during Bionano Genomics user meetings. The authors declare no other competing interests.
