## [Transparent Peer Review File · Communications Biology]

This manuscript has been previously reviewed at another Nature Research journal. This document only contains reviewer comments and rebuttal letters for versions considered at Communications Biology.

REVIEWERS' COMMENTS:

Reviewer #1 (Remarks to the Author):

I wish to thank the authors for satisfactorily addressing all of my previous comments and questions.

Reviewer #3 (Remarks to the Author):

The manuscript by Belser et al. (transferred and revised version) is of high quality and describes the successful integration of BNG optical maps with ONT long-read DNA sequencing data for banana as an example. The authors analyse double haploid and genetically "easy" material that does not suffer from significant complications. In general, this reviewer is satisfied with the improvements made in response to the points raised for the earlier version. However, there are a few open points as detailed below:

(Unfortunately, the PDF does not include line numbers. Even page numbers are missing.)

1) The wording to distinguish "genome" from "genome sequence" has been improved, but there are still problematic cases left. Example: One can not assemble a chromosome from sequence read data. The result of an assembly process is a chromosome sequence (or a pseudochromosome).

2) The fact the double haploid material has been used for sequencing is now mentioned in a better way, but the conclusions with regard to the "great" (according to the opinion of the authors) result of the paper are still stated exactly the same throughout the manuscript. Even if one takes into account that "selling results" is part of modern science, a bit more self-criticism would have been beneficial.

3) The extended section "Validation of telomere-to-telomere chromosomes" is a significant improvement of the manuscript. The new text includes this statement: "secondly the error rate of the nanopore technology is still too important to be corrected using polishing algorithms."

First: "important" is the wrong word. Please rephrase.

Second: This reviewer would argue that

(i) the coverage of the data generated by ONT is too low for high-quality polishing; and
(ii) that using short reads for polishing may reduce the error rate in CDS regions where short reads usually map easily, but it may "correct" towards a consensus sequence in (highly) repetitive regions. In other words, polishing with short reads may introduce errors compared to an ideal (fully correct) sequence.

4) Regarding the section on "Comparison of A, B and S-genomes"

(i) The authors mainly compare the genome sequences, not biological aspects of the genomes themselves. Once more, imprecise terminology.

(ii) There are biased statements that associate the sequencing technology used with the resulting sequence assembly. As pointed out (see Reviewer 2 for the first version), the data have been generated at different times. The bias in the statements needs to be removed. It is sufficient to address the various assemblies with the respective sub-genome letter.

Also, if the authors want to discuss a comparison of sequencing technologies, this needs to be done in the Discussion section and not in Results.

(iii) The authors should cite:

Comparison of the two up-to-date sequencing technologies for genome assembly: HiFi reads of Pacific Biosciences Sequel II system and ultralong reads of Oxford Nanopore (Lang et al, 2020, GigaScience).

Minor points:

- "Musa Schizocarpa" or "Musa schizocarpa"?

- 50Kb oder 50 Kb (space to separate value and dimension)? Anyway, this reviewer would prefer Mbp,

kbp, bp.

- Lack of punctuation makes reading difficult in some cases. Example:
"... were 469 Mbp and 474 Mbp in length, respectively, and ..."

Reviewer #1 (Remarks to the Author):

I wish to thank the authors for satisfactorily addressing all of my previous comments and questions.

We would like to thank the reviewer for reviewing our manuscript and with his advice we believe the manuscript is now of better quality.

Reviewer #3 (Remarks to the Author):

The manuscript by Belser et al. (transferred and revised version) is of high quality and describes the successful integration of BNG optical maps with ONT long-read DNA sequencing data for banana as an example. The authors analyse double haploid and genetically "easy" material that does not suffer from significant complications. In general, this reviewer is satisfied with the improvements made in response to the points raised for the earlier version. However, there are a few open points as detailed below:

(Unfortunately, the PDF does not include line numbers. Even page numbers are missing.)

1) The wording to distinguish "genome" from "genome sequence" has been improved, but there are still problematic cases left. Example: One can not assemble a chromosome from sequence read data. The result of an assembly process is a chromosome sequence (or a pseudochromosome).

We agree with the reviewer that we should almost always refer to "chromosome sequence" or "genome sequence" but this was a deliberate abuse of language in order to make the manuscript easier to read. We have made some adjustments and we hope they will be sufficient.

2) The fact the double haploid material has been used for sequencing is now mentioned in a better way, but the conclusions with regard to the "great" (according to the opinion of the authors) result of the paper are still stated exactly the same throughout the manuscript. Even if one takes into account that "selling results" is part of modern science, a bit more self-criticism would have been beneficial.

As recommended by the reviewer, we modified the first version of the manuscript to emphasize that we used a double haploid material. We did not find the word "great" throughout the manuscript, and only highlighted the contiguity and quality of the assembly which was generated using available software, and should be reproducible by the community. We never wanted to 'sell results' but only to show that using current technologies and open source software, it is now possible for a large haploid genome to obtain complete chromosome sequences.

3) The extended section "Validation of telomere-to-telomere chromosomes" is a significant improvement of the manuscript. The new text includes this statement: "secondly the error rate of the nanopore technology is still too important to be corrected using polishing algorithms."

First: "important" is the wrong word. Please rephrase.

We replace "important" by "high".

Second: This reviewer would argue that

(i) the coverage of the data generated by ONT is too low for high-quality polishing; and

(ii) that using short reads for polishing may reduce the error rate in CDS regions where short reads usually map easily, but it may "correct" towards a consensus sequence in (highly) repetitive regions. In other words, polishing with short reads may introduce errors compared to an ideal (fully correct) sequence.

We thank the reviewer for his advice, but we believe that a genome coverage of near 100X with long-reads is not a "low" coverage, and adding coverage will only slightly impact the quality of consensus (at least with R.9.4.1 nanopore flowcells, see 1). Regarding the easier correction of coding regions, we agree with the reviewer and this has already been pointed out in the revised manuscript. However, we have changed the sentence to be clearer, and have replaced "such as repetitive regions that are more difficult to polish" by "such as repetitive regions that are more difficult to polish and can generally contain more errors in nanopore assemblies".

1. <https://nanoporetech.github.io/medaka/benchmarks.html>

4) Regarding the section on "Comparison of A, B and S-genomes"

(i) The authors mainly compare the genome sequences, not biological aspects of the genomes themselves. Once more, imprecise terminology.

We modified the title of the section which is now "Comparison of A, B and S-genome assemblies"

(ii) There are biased statements that associate the sequencing technology used with the resulting sequence assembly. As pointed out (see Reviewer 2 for the first version), the data have been generated at different times. The bias in the statements needs to be removed. It is sufficient to address the various assemblies with the respective sub-genome letter.

Also, if the authors want to discuss a comparison of sequencing technologies, this needs to be done in the Discussion section and not in Results.

We replaced the bias statement "As a consequence that chromosome sequences of the A and S genomes contain less gaps" by the following sentence which only report facts "In addition, we noticed that chromosome sequences of the A and S genomes contain less gaps" and removed the following one "underlying the importance of ultra long reads to resolve these highly repetitive regions". As recommended we added a sentence in the Discussion section.

(iii) The authors should cite:

Comparison of the two up-to-date sequencing technologies for genome assembly: HiFi reads of Pacific Biosciences Sequel II system and ultralong reads of Oxford Nanopore (Lang et al, 2020, GigaScience).

We thank the reviewer for pointing out this reference, in which they found similar results, i.e. long-reads lead to better contiguity, we added the reference in the section comparing genomic sequences of A, B and S.

Minor points:

- "Musa Schizocarpa" or "Musa schizocarpa"?

We replaced the uppercase letter S with a lowercase one.

- 50Kb oder 50 Kb (space to separate value and dimension)? Anyway, this reviewer would prefer Mbp, kbp, bp.

We replaced the dimension as recommended and added a space to separate value and dimension.

- Lack of punctuation makes reading difficult in some cases. Example:

"... were 469 Mbp and 474 Mbp in length, respectively, and ..."

We made the edits as recommended.